# Metabolic dysregulation in vitamin E and carnitine shuttle energy mechanisms associate with human frailty

Nicholas J.W. Rattray [1,2,3]*, Drupad K. Trivedi[1], Yun Xu [1,4], Tarani Chandola[5], Caroline H. Johnson[2], Alan D. Marshall[5,8], Krisztina Mekli[5], Zahra Rattray [3], Gindo Tampubolon[5], Bram Vanhoutte [5,9], Iain R. White[1,10], Frederick C.W. Wu[6], Neil Pendleton [7,11], James Nazroo[5,11] & Royston Goodacre[1,4,11]

Global ageing poses a substantial economic burden on health and social care costs. Enabling a greater proportion of older people to stay healthy for longer is key to the future sustainability of health, social and economic policy. Frailty and associated decrease in resilience plays a central role in poor health in later life. In this study, we present a population level assessment of the metabolic phenotype associated with frailty. Analysis of serum from 1191 older individuals (aged between 56 and 84 years old) and subsequent longitudinal validation (on 786 subjects) was carried out using liquid and gas chromatography-mass spectrometry metabolomics and stratified across a frailty index designed to quantitatively summarize vulnerability. Through multivariate regression and network modelling and mROC modeling we identified 12 significant metabolites (including three tocotrienols and six carnitines) that differentiate frail and non-frail phenotypes. Our study provides evidence that the dysregulation of carnitine shuttle and vitamin E pathways play a role in the risk of frailty.

[1] School of Chemistry, Manchester Institute for Biotechnology, University of Manchester, 131 Princess Street, Manchester M1 7DN, UK. [2] Department of Environmental Health Sciences, Yale School of Public Health, Yale University, 60 College Street, New Haven, CT 06520, USA. [3] Strathclyde Institute of Pharmacy and Biomedical Sciences, University of Strathclyde, 161 Cathedral Street, Glasgow G4 0RE, UK. [4] Department of Biochemistry, Institute of Integrative Biology, University of Liverpool, Biosciences Building, Crown Street, Liverpool L69 7ZB, UK. [5] Cathie Marsh Institute for Social Research, University of Manchester, Manchester M13 9PT, UK. [6] Faculty of Medical and Human Sciences, Institute of Human Development, Centre for Endocrinology and Diabetes, Andrology Research Unit, Manchester Academic Health Sciences Centre, University of Manchester, Manchester, UK. [7] Division of Neuroscience and Experimental Psychology, School of Biological Sciences, Manchester Academic Health Science Centre, University of Manchester Salford Royal Hospital, Manchester M6 8HD, UK. [8] Present address: School of Social and Political Science, University of Edinburgh, Chrystal Macmillan Building, 15a George Square, Edinburgh EH8 9LD, UK. [9] Present address: Department of Sociology, Eleanor Rathbone Building, University of Liverpool, Bedford Street South, University of Liverpool, Liverpool L69 7ZA, UK. [10] Present address: Laboratory for Environmental and Life Sciences, University of Nova Gorica, Vipavska 13, SI-5000 Nova Gorica, Slovenia. [11] These authors contributed equally: Neil Pendleton, James Nazroo, Royston Goodacre.  *email: nicholas.rattray@strath.ac.uk

A consequence of ageing is the decline in biological function that will eventually lead to a progressive deterioration of physiological performance, a decline in the ability to respond to stress and an associated increase in vulnerability. Since 1950 life expectancy has been rising at a rate of more than three years per decade, and since the onset of the millennium this has risen to five years per decade[1]. As a consequence, it is estimated that between 2015 and 2050 the global proportion of over 60-year olds will increase from 12 to 22%[2], while the number of over-65s is forecast to triple by 2050[3], even though recent evidence suggests that maximum lifespan could be fixed[4,5].

While increasing life expectancies are a positive development, all individuals are still currently subject to natural constraints and face decline in health status with age (albeit at rates that vary across populations and sub-populations[6]) increasing the risk of exposure to chronic age-related disorders[7]. So, in 2015 the global healthy life expectancy at birth (HALE) was calculated to be 63.1 years[8], which indicates a substantial burden of later life morbidity, even though rises in HALE are a testament to continuing global healthcare advancements including medical diagnosis and treatment of cardiovascular disease[9], immunization[10], smoking cessation[11], healthy diet[12] and an increased understanding of social determinants of health[13]. Such demographic shifts and resulting population ageing are leading public health practitioners and policy makers to actively push for global innovations that can positively shape the health of future elderly populations[14–16]. Indeed, healthy ageing in later life has enormous potential to affect society as a whole. Consequently, to mitigate the potential economic and social strains that flow from population ageing brings call for direct government initiatives to develop appropriate public health interventions to reduce the impact of functional decline associated with frailty.

Clinically, the effects of frailty are defined as a multimodal syndrome emphasized by a loss of internal reserves (energy, physical ability, balance, cognition, and health) that gives rise to biological vulnerability within an individual. This inherent complexity means that there is no single diagnostic tool available to identify the presence and extent of frailty. However, a myriad of scoring instruments are reported within the literature. These typically assess a number of negative outcomes, or distal phenotypes, and are validated by stratifying the scoring aspects of large patient cohorts such as surgical prognosis[15], primary care interventions[16–20] and associated key detrimental events and are subsequently used in the development of instruments that can be used to evaluate the presence of frailty, such as the frailty phenotype[21] and frailty index (FI)[22]. Although these methods have acceptable performance in identification of frailty status, they have some limitations, including subjectivity of individual answers, resource utilisation costs and lack of linkage to underlying biological mechanisms[23].

Orthogonal to such procedures, biochemical assessments of frailty in the context of biomarker detection are sought after as they hold the potential to identify biological pathways that contribute to a frailty phenotype, offering opportunities in developing strategies for the identification and management of frailty. A focus for the bio-gerontology community within this area has been the chemistry of life contained within the central dogma of molecular biology. Several large cohort studies have demonstrated correlation between mitochondrial DNA and energy co-factors alongside mortality[24,25], dysregulation of transcriptional networks and age-dependent decline[26], alongside metabolic signatures of biological ageing in young[27] and older people[28]. By contrast, no such large scale, population level comparison of frailty identified using a validated measure versus metabolism has been undertaken. Yet the ability to uncover downstream biochemical relationships by applying metabolomics based approaches

hold great potential for the development of public health strategies to reduce the risk of frailty and its adverse consequences.

As the complex links between age-related frailty and the underlying life-course, social, psychological, genetic and metabolic processes remain unclear, the fRaill project (www.micra.manchester.ac.uk/research/fraill/) takes an interdisciplinary approach to examine the causal processes relating to frailty and wellbeing at older ages. Within this work, we performed high-throughput untargeted mass spectrometry-based metabolic profiling coupled with pathway and network analysis on longitudinal serum samples, taken four years apart, from a cohort of well-phenotyped ageing (≥58 years old) subjects from the English Longitudinal Study of Ageing (http://www.elsa-project.ac.uk/). The aim of this approach is to stratify metabolic phenotypes (metabotypes[29]) over a FI derived from over 60 measured indicators and assess whether associated biochemical networks relate to biological degeneration associated with frailty. From this work we have identified a panel of 12 metabolites associated with the FI and from these, two chemical classes (tocotrienols and carnitines) exhibit significantly modulated under-expression when overlaid across the FI. Subsequent network enrichment analysis and statistical modelling has identified carnitine shuttle and vitamin E metabolism as two modulated pathways that are related to a higher energy metabolic phenotype. The ability of the combined metabolic model to predict frailty has also been confirmed by a cross validated ROC model. In conjunction with these results, Mendelian Randomization analysis has been performed on previously collected GWAS data to determine the causal relationship of frailty to carnitine levels. This indicates a level of significant association between decreased levels of carnitine and frailty.

## Results

**Frailty as a function of ageing.** In the first stage of our analysis we performed cross-sectional stratification of a continuous FI over mass spectrometry-derived metabolite data within the fRaill project. The aim was to identify underlying chemical determinants of frailty (or resilience) and build an associated metabolic pathway model. Sample members participated in a face-to-face interview, a nurse assessment of physical function, anthropometric measurements and collection of blood samples. A full list of questionnaire characteristics and summary metadata can be found here—http://www.elsa-project.ac.uk/data_elsa. A FI model was developed by calculating a cumulative score for each individual using the presence/absence of 60 deficit variables contained within Wave 4 ($n = 1846$) and Wave 6 ($n = 1753$) of the ELSA (English Longitudinal Study of Ageing; following standard practice, sample members were included if they had responses to at least 30 of these items)[6]. These items covered a broad range of attributes such as cognitive function, falls and fractures, vision, hearing, chronic diseases and depression. Due to the complexity in index-coding and cut-off point determination, the presence of each item attributed varying predetermined amounts to an individual's FI score[30] which had an overall range of 0.04–0.698 over the full sample cohort. Subsequent plotting of the FI distribution produced a unimodal right skewed distribution of data from both waves (Supplementary Fig. 1). This distribution of frailty scores is consistent with findings from other population studies[22]. To deal with this skewed distribution we subsequently stratified the FI scores into four categories, <0.1 (26.2%), 0.1–0.2 (48.4%), 0.2–0.3 (17.5%), >0.3 (7.9%).

Previous studies have indicated that frailty increases as subjects age, but this is more closely linked to biological age rather than chronological age[31]. To investigate this, a robust linear regression (RLR) model was developed to determine the level of correlation

**a**

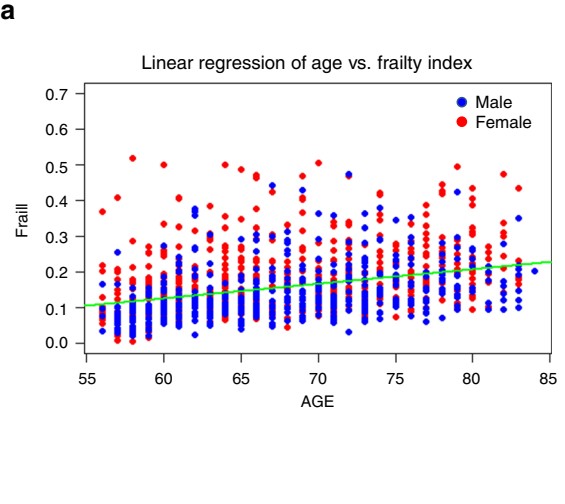

**b**

| Attribute at Wave 4 of ELSA ($n = 1191$) | Mean | SD |
|---|---|---|
| Age | 67.7 | 7.15 |
| % female (690F/501M) | 57.9 | N/A |
| Rockwood Frailty Index | 0.16 | 0.09 |
| BMI | 27.8 | 5.0 |
| Blood fibrinogen level (g/l) | 3.35 | 0.54 |
| Total blood cholesterol level (mmol /l) | 5.58 | 1.22 |
| Blood hdl level (mmol /l) (high density lipoprotein) | 1.57 | 0.40 |
| Blood triglyceride level (mmol /l) | 1.65 | 0.94 |
| Blood ldl level (mmol /l) (low density lipoprotein) | 3.27 | 1.06 |
| Blood ferritin level (ng/ml ) | 118.55 | 105.81 |
| Blood crp level (mg/l) | 3.54 | 7.09 |
| Blood dehydroepiandrosterone (dheas) level (umol /l) | 2.20 | 1.65 |
| Blood insulin-like growth factor (igf-1) level (nmol /l) | 15.82 | 5.59 |
| Blood glucose level (mmol /l) - fasting samples only | 4.93 | 0.81 |
| Blood haemoglobin level (g/dl) | 14.02 | 1.22 |
| Blood glycated haemoglobin level (%) | 5.88 | 0.70 |
| White blood cell count ($\times 10^9$ cells/litre) | 6.18 | 1.85 |
| Blood mean corpuscular haemoglobin level (pg/cell) | 30.38 | 2.26 |

**c**

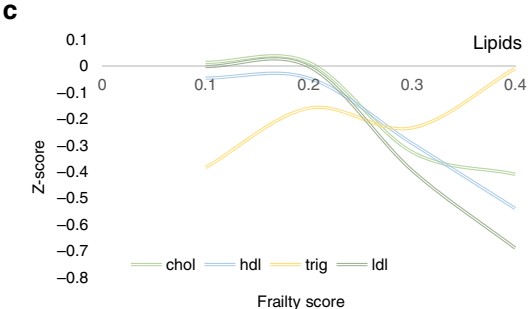

**d**

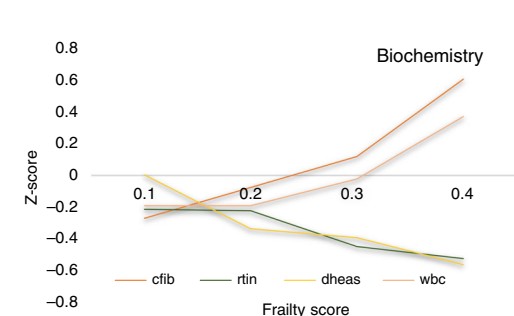

**Fig. 1** Sample attributes at wave 4 of ELSA. **a** Linear regression of Age vs. Frailty Index score indicating a moderate correlation and implying that the concept of frailty, when measured under the Rockwood FI scoring system, is an independent variable with respect to age (blue dots = male subjects, red dots = female subjects). **b** Mean sample characteristics from 1191 subjects and associated blood analysis. **c** Mean cholesterol levels as observed across the frailty distribution using the standard scoring method. It can be seen that LDL, HDL and cholesterol all decrease when entering the non-frail cut-off[50]. Triglycerides are seen to increase. **d** Most pronounced biochemistry levels as observed across the frailty distribution using the standard scoring method. Fibrinogen and white blood cells indicating a marked increased Z-score over the frailty distribution whereas ferritin and dehydroepiandrosterone indicate a decrease. Source data are provided as a Source Data file

between FI and age (Fig. 1a, Supplementary Figs. 2–7). The regression equation for the full FI (in the form $y = mx + c$) is equal to FI $= 0.119x$Age $+ 0.004092$. Derived from this the SSE (sum of squares of error) $= 5.7017$ and *R squared* (coefficient of determination) $= 0.3357$, where $1 =$ total correlation and $0 =$ non-correlation. As a measure of the discrepancy between the observed data and the estimation made by the linear regression model, the *R* value of 0.34 explains a reasonable proportion of variance in frailty explained by age which is to be expected given the measure over a large age range. But, the sum-of-square-error of estimates (SSE) still indicates there is still a large amount of unexplained variance that is present across all ages. Similar models for each FI category were also calculated (Supplementary Fig 2) indicating a drop in the correlation between age and FI as FI value increases. These results indicate that although age is linked to the frailty phenotype, as a subject expresses a stronger frailty phenotype, age plays a lesser role in the classification scoring.

**Cross sectional modelling of biochemistry and metabolism over the frail index.** Untargeted metabolic phenotyping was performed on serum samples collected at Wave 4 of the ELSA alongside a range of standard biochemistry assays (see Supplementary Methods). 1846 subjects were used to generate the FI and a combined total of 1191 serum samples were selected for untargeted multiplatform metabolomics analysis based on availability, quality of sample and metadata inclusiveness. Within the complete sample set 57.9% ($n = 690$) were women and other major characteristics are represented in Fig. 1b. Age and sex dependent changes in biochemical measurements of lipids (LDL, HDL, cholesterol and triglycerides) alongside other blood constituents (white blood cell count, dehydroepiandrosterone, fibrinogen and ferritin) were calculated using a standard scoring method (z-score, giving a standardised score with a mean of zero and a standard deviation of one (also referred to as auto-scaling)). This methodology required a complete set of input data items for all subjects to perform the analysis, but 428 subjects had at least one data point missing (The percentage of missing biomarkers variables varied between 0 and 2.5% except fasting glucose which has displayed 32% missings out of 1196 observations). To account for this, we adopted a missing value imputation approach (multivariate multiple imputation method with known seed for replication[32]) to enable the assessment of all data point for all subjects. This completed dataset was subsequently tested by further sensitivity analysis and all biochemistry measures were then re-stratified over the FI as a whole and stratified in to male and female subgroups. Pearson's correlation scores were calculated for non-standardized biomarker values and only significantly correlated biomarkers were used for standardisation purposes. Individuals were then grouped into four classes based on FI and at the same time, male vs. female stratification was also performed (data in Supplementary Figs. 8–11). Subsequent plots indicate ±correlations over the FI (Fig. 1c, d). Using the FI score of each subject

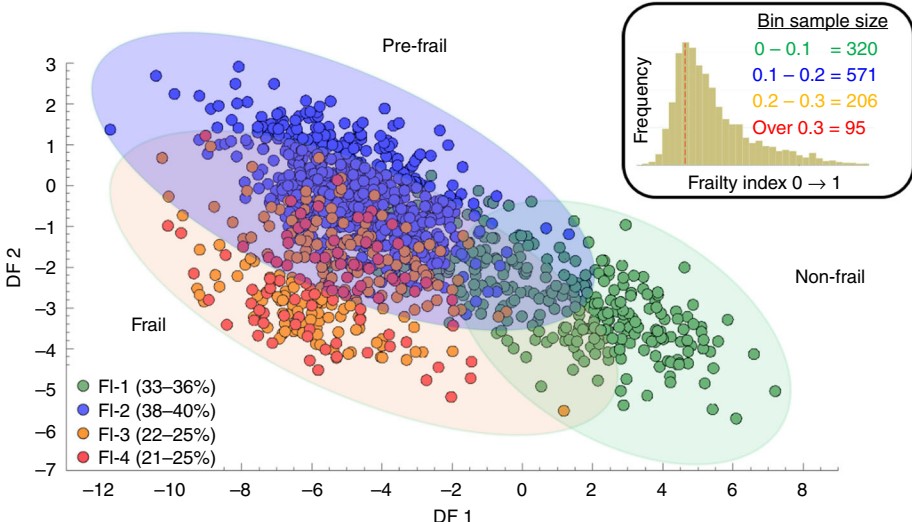

**Fig. 2** PC-DFA of serum metabolite data stratified over the frailty index distribution. Principal component discriminant function analysis (PC-DFA) carried out on serum metabolite profiling data from 1191 subjects within wave 4 of ELSA. Data were $\log_2$ transformed and stratified in to groups determined by Rockwood Frailty Index value. The results were cross validated by bootstrapping (10,000 iterations) and indicate two clear planes of separation along the 0.1–0.2 axis and the 0.2–0.3 axis. This data correlates with observations that directly stratify clinical assessment over the frailty index indicating three distinct clinical phenotypes[55]. Green circle = Frailty Index 0–0.1, Blue circle—Frailty Index 0.1–0.2, Orange circle = Frailty Index 0.2–0.3, Red circle = Frailty Index above 0.3). Source data are provided as a Source Data file

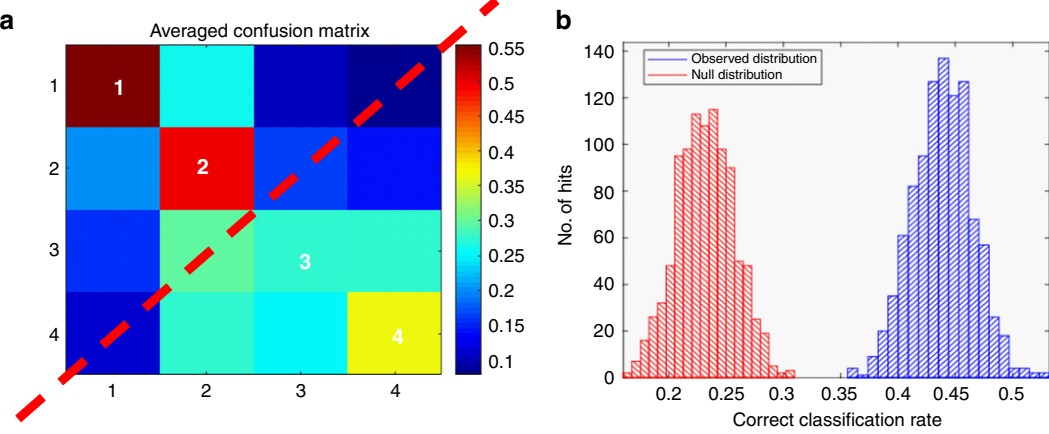

**Fig. 3** RUSBoost-CART analysis of samples binned over the frailty index. **a** Machine learning based Random Under Sampling boosting Classification and Regression Tree analyses on (+) mode UHPLC-MS data supporting correct sample stratification over frailty index distribution. Confusion Matrix indicates a clear separation between >0.2 and <0.2 on the frailty index and thus good model prediction. **b** Null-distribution classification rate (red frequency histogram) supporting machine learning results (blue frequency histogram) and, indicating the groupings in the confusion matrix are correctly classified. Source data are provided as a Source Data file

as a supervisory variable (*y*-output) principal component-discriminant function analysis (PC-DFA) modelling allowed for the stratification of the mass spectrometry metabolite data (*x*-input) over the FI. This PC-DFA approach was used to cluster the UHPLC-MS and GC-MS datasets and the scores plot (Fig. 2) reveals the relationship between the four FI classes and indicates that two clear channels of separation along the 0.1–0.2 and 0.2–0.3 FI axis. Inspection of the loadings vectors also allows for the investigation of the presence of metabolic differentiation between these four levels.

The development of the combined PC-DFA and Random Under Sampling boosting Classification and Regression Tree analyses (RUSBoost-CART) models were performed to reduce the dimensionality of the multivariate LCMS and GCMS datasets while simultaneously investigating the presence of metabolic differentiation between the four levels of scoring within the frail index (Fig. 3). The RUSBoost-CART model was double cross validation (2CV) validated using resampling methods of bootstrapping ($n = 10,000$) on the training set only and permutation testing was used to generate null distributions[33]. This method indicated a strong separation between >0.2 and <0.2 on the FI and a moderate level separation between <0.1 and 0.1–0.2, thus supporting similar clustering within the PC-DFA. This was also supported by the subsequent null-distribution classification rate. Subsequent univariate analysis was carried out to determine the statistical significance of individual metabolites modulated by the frailty metabotype. Non-parametric t-tests cross-validated by false discovery rates were used to assess metabolite significance between samples that lay at <0.2 and >0.2 on the FI. Spearman based correlation was also performed to determine between

metabolite association and develop associated clusters as indicated by the heatmap in Supplementary Fig. 12.

**Development of a metabolic network of frailty.** Having established the presence of separation in FI level using the PC-DFA approach, we subsequently used the mummichog based pathway enrichment enrichment[34] to predict network activity and identify biochemical pathways modulated by the frailty metabotype. This method has successfully been applied to a diverse range of clinical areas such as liver damage[35,36] and T cell activation[34,37]. The general analysis pipeline of using XCMS deconvolution in conjunction with the mummichog pathway method to develop an integrated systems approach has already been documented[38]. By applying this methodology and mapping *m/z* clustering differences on to an integrated metabolic network containing data from the UCSD BiGG[39], KEGG http://www.genome.jp/kegg/kegg1.html and Edinburgh Human Metabolic Network[40] resources, metabolic differences between frail versus non-frail sample classes were identified. The generation of subsequent metabolic activity networks highlighted in Fig. 4 (and Supplementary Fig. 13) identified 25 metabolites present within four metabolic pathways (the carnitine shuttle, peroxisomal degradation, the kynurenine

pathway and vitamin E metabolism) and were statistically significant in the transition from non-frail to frail metabotypes. From these 25 metabolites, 12 were calculated as individually statistically significant (using a FDR corrected Mann–Whitney test) in distinguishing frailty class and used to generate a multivariate ROC prediction model (Fig. 5).

**Biological validation of frailty metabolic phenotype via longitudinal analysis.** To confirm the metabolic dysregulation observed within the cross-sectional model, longitudinal analysis on samples from wave 6 of ELSA (samples from the same subjects taken four years later) was undertaken to provide biological validation of the significant metabolites identified. A FI was calculated from metadata belonging to the 1753 subjects (238 lower than Wave 4 due to subject attrition) and as before a unimodal right skewed distribution of data was observed (Supplementary Fig. 1). Subsequently, 786 serum samples were retrieved and untargeted UHPLCMS metabolic phenotyping was performed to determine if the same metabolites were present with corresponding non-frail to frail directionality (430 samples lower than Wave 4 due to subject attrition and sample availability). Eleven metabolites from wave 4 with similar non-frail to frail trajectories

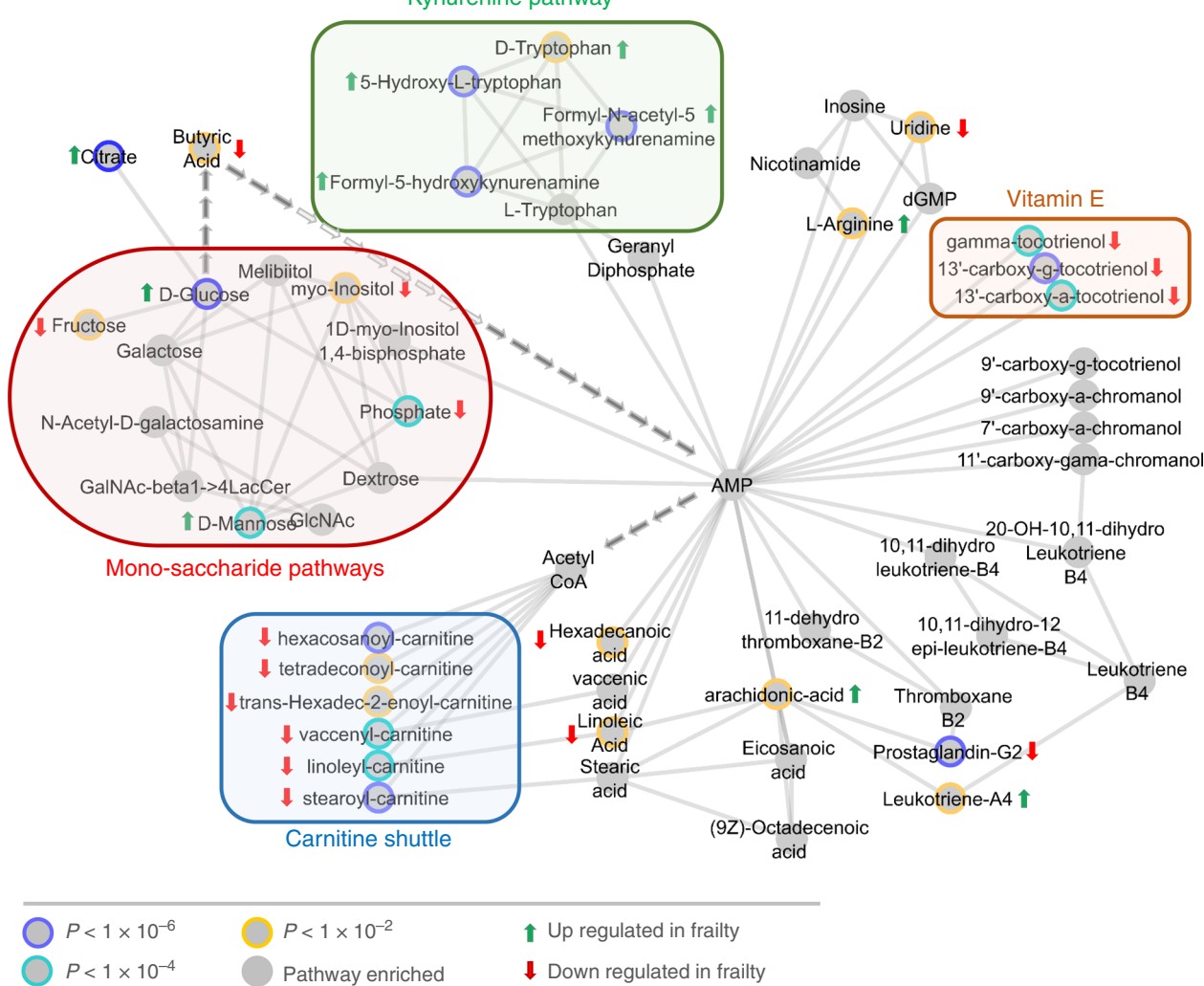

**Fig. 4** Enriched pathway model from hybrid network analysis. Frailty metabolite subnetwork generated from the human metabolite network from within the mummichog-Cytoscape pipeline using 554 metabolite features with unique *m/z* values from the LCMS (+) analysis alongside the addition of 86 metabolites from GCMS analysis. This combined approach highlights 4 main metabolic areas that altered within the frailty metabotype, all of them identifying cyclic AMP as a potential hub-metabolite. Source data are provided as a Source Data file

**a**

| Metabolite | t.stat | p.value | FDR | ROC AUC |
|---|---|---|---|---|
| Prostaglandin G2 | −5.4509 | 6.09E−08 | 1.40E−06 | 0.611 |
| Formyl-N-acetyl-5-methoxykynurenamine | 5.0241 | 5.83E−07 | 6.71E−06 | 0.609 |
| Subs-eicosadienoicacid | −4.7961 | 1.82E−06 | 1.40E−05 | n/a |
| 5-Hydroxy-L-tryptophan | 4.355 | 1.45E−05 | 8.31E−05 | n/a |
| Formyl-5-hydroxykynurenamine | 4.2737 | 2.08E−05 | 9.55E−05 | n/a |
| Urocortisol | 3.4211 | 0.000645 | 0.002472 | 0.584 |
| Hexacosanoylcarnitine | −3.23 | 0.001272 | 0.004179 | 0.626 |
| Stearoylcarnitine | −3.1527 | 0.001658 | 0.004768 | 0.651 |
| Linoleylcarnitine | −2.8412 | 0.004571 | 0.011681 | 0.622 |
| 13′-carboxy-alpha-tocotrienol | −2.7019 | 0.006993 | 0.016084 | 0.648 |
| Fructose | 2.5038 | 0.01242 | 0.02597 | n/a |
| 13′-carboxy-gama-tocotrienol | −2.3854 | 0.017219 | 0.031802 | 0.63 |
| L-Arginine | 2.3694 | 0.017975 | 0.031802 | 0.595 |
| Gama-tocotrienol | −2.2982 | 0.021725 | 0.035691 | 0.624 |
| Leukotriene A4 | 2.2258 | 0.026217 | 0.0402 | n/a |
| Vaccenyl carnitine | −2.1636 | 0.030694 | 0.044123 | 0.616 |

**b**

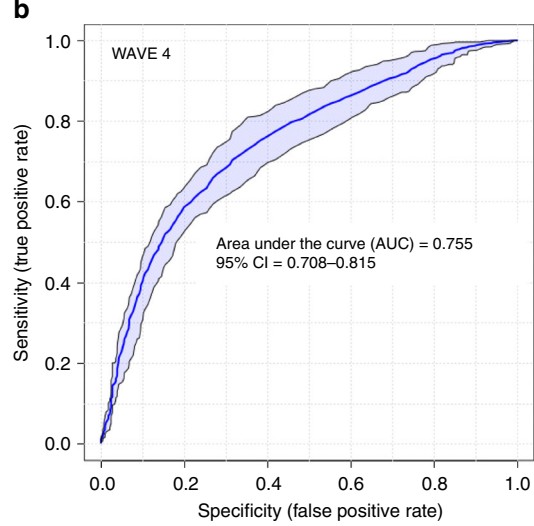

WAVE 4

Area under the curve (AUC) = 0.755
95% CI = 0.708–0.815

**c**

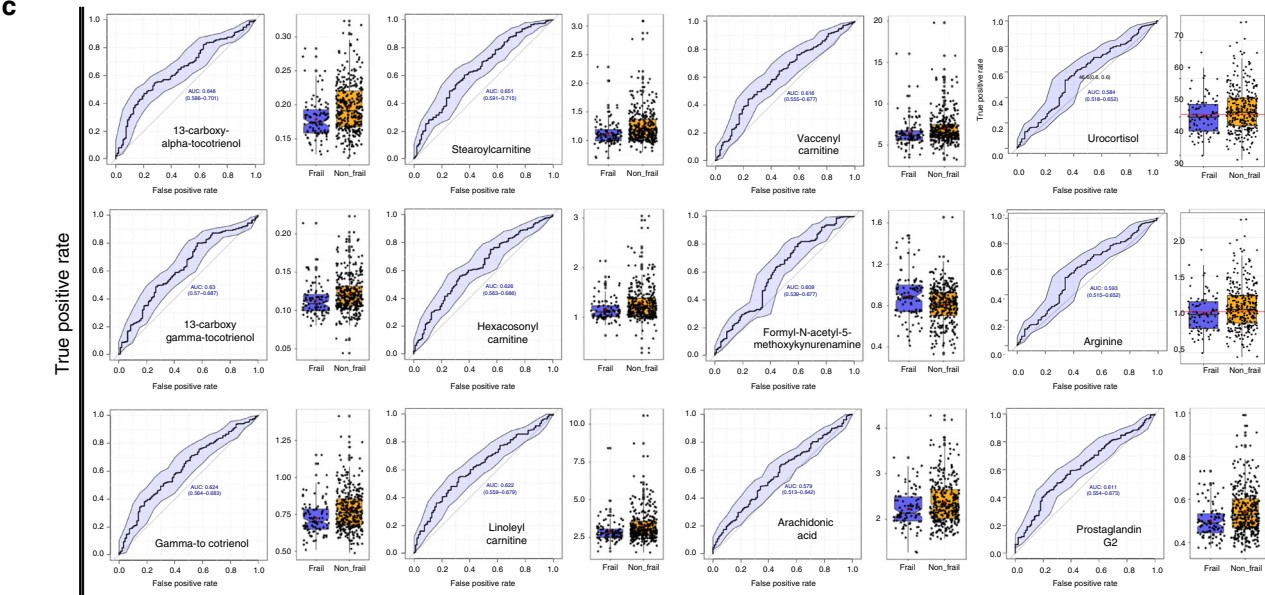

**Fig. 5** Significance and predictive ability within the metabolic model of frailty. **a** Table indicating 12 metabolites of statistical significance ($p < 0.05$) in differentiating non-frail and frail metabolic phenotypes using Kruskal-Wallis analysis of variance with subsequent false discovery rate testing for multiple comparisons. Data also contains area under curve data used to create multivariate receiver operating characteristic curve (mROC) **b** mROC curve from Waves 4 generated by combining 12 metabolites to generate a predictive model of frailty status. The shaded area indicate 95% confidence intervals calculated by Monte Carlo cross validation using balanced subsampling and 1000 iterations of bootstrapped cross-validation. **c** Univariate ROC curves and non-frail (orange) to frail (blue) boxplots of each metabolite used to generate the multivariate ROC analysis. Each boxplot displays a median value (centre line), upper and lower quartiles (box limits), 1.5× interquartile range (black bar), and points out of interquartile range are outliers. Source data are provided as a Source Data file

were identified at Metabolomics Standard Initiatives level 1 or 2 within the analysis[41] (commonly used metabolite identification protocol devised and used by the global metabolomics community), with major components from the carnitine shuttle and vitamin E pathways still present (Supplementary Table 1). The detected metabolites were used to replicate the multivariate ROC model within the cross-sectional analysis.

**Mendelian randomization analysis of carnitine levels and their influence on frailty**. To investigate the causal effect of carnitine levels on frailty we conducted Mendelian Randomization analysis[42]. We analysed three exposures related to increased/decreased carnitine levels in blood and used four SNPs as genetic

instruments (rs12356193, rs419291, rs1466788, rs1171606), derived from two studies[43,44]. We found that Odds Ratio of frailty per 10% decrease in carnitine was 1.53 (95% CI = 1.01–2.29, $p = 0.042$, genetic instruments: rs12356193 and rs419291, Inverse variance weighted method), providing significant evidence for the causal relationship.

**Investigation of correlation between model confounders and co-morbidities vs. frailty**. To investigate the effects of model adjustment fully, we performed a global analysis of a range of co-morbidities and biochemistry factors a using multidimensional scaling analysis approach. This tool was used to measure the level of similarity between factors that are closely correlated (or

anti-correlated) to the frailty indices. Through this method five different factors identified as being highly correlated to frailty were (Supplementary Fig. 14):

Age—Already discussed in the article as being correlated to frailty (using linear regression approach) but does not account for all variation.

• hsCRP—The pathogenic mechanisms of C-reactive protein (CRP) in ageing involves binding to FcγRII and activation of the TGF-β/Smad3 and non-TGF-β/Smad3 signalling pathways. These are directly and indirectly used to induce inflammation and fibrosis thus impairing the ability of a cell to proliferate and ultimately contributing to the process of ageing.

Cfib—Plasma fibrinogen levels were noted to increase over the FI and this relationship has been noted in ageing studies looking at both male and female subjects.

HbA1c/Fasting-glucose—Within the literature, there is a wealth of evidence that demonstrates HbA1c levels increase as non-diabetic subjects age. Our similar observation of the association of frailty to HbA1C fits well with this theory due to the high level of correlation to frailty to age.

## Discussion

At the biochemical level, ageing is a continuous and dynamic remodelling process of metabolism and cell function. This chemical reconditioning is heavily influenced by unrepaired accumulation of DNA mutational damage occurring within nuclear DNA[45] and mitochondrial DNA[46,47] brought about by environmental stressors. Ensuing dysfunction can be translated back to physiological status and contribute to an ageing phenotype. Indeed, population studies have already examined metabolic baseline levels in human health[48] and longevity[28], showing that metabolomics combined with symptom, biochemical or demographic data can successfully identify distinct biochemical models that were not previously been associated with lifespan in humans. These studies not only indicate modulation of various metabolic pathways such as those within the TCA cycle[28] and lipid biosynthesis[49,50], but also suggest that large sample sizes ($n > 600$)[48] and precise analytical methodologies, such as those performed within the HUSERMET study[48,51], are essential for robust analysis of the data generated. Yet, metabolic studies directly investigating frailty have previously focused primarily on the influence of specific disease states, such as breast cancer[52] ($n = 79$) and sarcopenia[53] ($n = 139$), and have not specifically analysed the broad underlying causal processes relating to the frailty-ageing condition. With the aim of expanding knowledge in this area, our goal was to identify the presence of a potential frailty metabolic phenotype and link it to associated physiological and pathophysiological processes. Using a validated assessment of frailty status in conjunction with standard biochemistry analysis and high-throughput metabolic profiling, we generated a metabolic network that highlights significant areas of metabolism that are associated with the clinically assessed FI. This multi-level approach developed a mROC model that identified 12 metabolites as being highly significant in the differentiation of subjects exhibiting frail and non-frail phenotypes as indicated by their position on the FI. Ultimately, our studies show that global lipid metabolism is changed under the frailty phenotype and down regulation of the carnitine shuttle and vitamin E metabolism show potential in playing a role in modulating cellular energy production. These biochemical observations in turn mirror the reduced state of physical activity observed clinically in frail subjects.

Initial calculations of the frailty indices used within the study generated unimodal right-skewed distributions[6] (Supplementary Fig. 1), comparable to those developed in other population scale assessments of frailty[30]. The operationalization of the Wave 4 index into a range of four discreet classifiers applicable to stratification over mass spectrometry based metabolomics data was achieved by binning FI scores in to four supervisory classes and applying PC-DFA (Fig. 2). Upon analysis, this approach identified only three distinct sub-planes of separation along the 0.1–0.2 axis and the 0.2–0.3 axes of FI scoring. These metabolic level observations correlate with independent FI assessments made in other large scale studies of frailty across the globe, such as in Canada[18,54,55] and Taiwan[51] in which non-frail, pre-frail and frail discreet classifiers were considered to have equivalent FI scores. Subsequent whole index validation by PLS methodologies (Fig. 3) and individual bin assessment using linear regression (Supplementary Fig. 2) also indicated that correlation between age and frailty actually decreases across the index thus distinguishing it from normal age-related degeneration, further supplying validation to the concept that frailty is in fact a geriatric syndrome within its own right and, although influenced by age, distinct from normal temporal changes.

Prior to metabolomics and pathway analysis, a panel of standard clinical biochemical tests were performed on matched blood samples to investigate how conventional assays, already routinely used within clinical practise, could be used to assess and develop the frailty phenotype. Stable cholesterol, LDL and HDL levels were noted within the non- and pre-frail phenotypes, but sharp decreases were associated with the frail phenotype (Fig. 1c). These results are confirmed by previous experimental data in which serum cholesterol levels have been indicated as a hematologic marker of frailty in older hospitalized patients[56,57]. LDL/HDL levels have also been demonstrated to decrease with age[58], and conversely, high levels of HDL has also been directly associated with better survival rates in very old subjects[59]. However, a fluctuation in triglyceride levels was observed across the FI range. Within these experimental studies weight loss is identified as the key explanatory variable, which parallels the importance of involuntary weight loss displayed within the frailty phenotype.

Steadily increasing fibrinogen and white blood cell levels were also noted across the FI (Fig. 1d). Fibrinogen, as an essential component of the coagulation cascade and a key regulator of inflammation, which has been implicated as a risk factor for several diseases[60], the elevation of which, has previously been associated with increasing frailty level[61]. In the present study observed white blood cell levels were directly correlated with frailty in older adults, an observation that further supplies evidence for the role of immuno-endocrine cross-talk within[62] functional decline. Serum ferritin levels were also noted to decrease over the FI which would initially infer an increase in anaemia. However, previous studies designed to investigate the utility of ferritin as a single indicator of frailty determined it to be of exceedingly low potential[63] owing to the complex interactions between serum iron, total iron binding capacity and transferrin saturation ratio severely hampering levels of assay sensitivity. To compound the use of serum ferritin as a biomarker of frailty further, increased levels are associated with an increase in oxidative stress and cellular damage[64] which goes against observed values obtained within this study. Dehydroepiandrosterone (DHEAS) levels were also evaluated over the index range, the decline in which was correlated with a higher FI. This correlation is in agreement with previous studies reporting a widely-recognised association between decreasing androgen levels and ageing[65,66].

All clinical biochemistry data were also analysed controlling for sex. As a result, an interesting observation was noted within the measured triglyceride levels. Upon stratification, male vs. female triglyceride levels act in a divergent manner (see Supplementary Fig. 10) as FI score increases; females noting a sharp increase and

males noting a decrease. The identification of this important role of triglyceride levels has already been documented in the Leiden Longevity Study ($n = 1664$)[63] in which multiple regression models indicated decreased triglyceride levels predicted to serve as an indicator of longevity in females.

Biochemical network activity assessment, in which all *m/z* features were used as input, detected 25 identified metabolites (Supplementary Table 1) that contributed to dysregulation of four metabolic pathways (Fig. 4)—monosaccharide, kynurenine, vitamin E and carnitine metabolism. All individual pathways contain a link to energy production within eukaryotic cells. In this process, pathways identified as significant can contain individual metabolites that may not be significant on their own—due to their presence within a pathway that has other significant features contained within it. To investigate the role of the individual 25 metabolites in differentiating non-frail and frail metabolic phenotypes, Kruskal-Wallis analysis of variance with subsequent false discovery rate (FDR) testing for multiple comparisons was used to test for significance. In total 12 metabolites (Fig. 5a) were deemed individually statistically significant (>0.05) in differentiating non-frail and frail metabotypes. These feature were then used to develop a Multivariate Receiver Operating Characteristic (mROC) curve (Fig. 5b), to act as a predictive model of frail status. In this process, the final mROC model used 12 metabolites (individual distributions and univariate contributing ROCs shown in Fig. 5c) to generate an AUC of 0.755 (95% CI = 0.708–0.815)—indicating a moderately strong level of performance. Overall results from combining PC-DFA separation, RUSBoost-CART sampling validation, pathway enrichment, univariate descriptive comparison of metabolite means and concluding mROC predictive modelling provide a diverse range of evidence that all support the theory of metabolic dysregulation within the frail metabotype. As further evidence, predictive modelling was also replicated on in a validation subset of 768 samples from the same subjects collected four years later. The presence of 9 out of 12 the metabolites used to generate the wave 4 mROC model were detected within the deconvolved Wave 6 mass spectrometry dataset. The data from these features was then used to generate a mROC model from the validation subset with an AUC of 0.702 (95% CI = 0.63–0.748) (Supplementary Fig. 15) indicating a reproducible result even with slightly reduced data input.

Using the two main pathways identified within pathway analysis, two of the four genetic instruments used in the Mendelian Randomization analysis showed evidence for the causal effect of carnitine levels in frailty. Our instrument SNPs represent the *SLC16A9* (solute carrier family 16 member 9) (rs12356193) and *SLC22A4* (solute carrier family 22 member 4) (rs419291) genes. These SNPs were strongly associated with carnitine levels in a study of human metabolites ($p = 3.69 \times 10^{-63}$ and $p = 3.1 \times 10^{-18}$, respectively)[43]. Although our results do not survive strict correction for multiple testing, they are firmly supported by the literature. A recent study measuring common variants (minor allele frequency >5%) using healthy ageing as outcome reported the possible involvement of the *SLC22A4* gene, represented by multiple variants, including rs419291[67].

Vitamin E analogues, in this case detected tocotrienols, are well-documented due to their lipoperoxyl radical-scavenging abilities in the termination of lipid peroxidation via proton transfer on to lipid free radicals[68]. However, they are also noted for their ability to scavenge reactive nitrogen species, inhibit cyclooxygenase- and 5-lipoxygenase-catalyzed eicosanoids, and suppress pro-inflammatory signalling, such as NF-κB[69]. This reduction of free radical-mediated oxidative damage alongside general inflammatory suppression is vital to the maintenance of a healthy lifestyle over time. Breakdown of the endogenous antioxidant system can lead to the accumulation of oxidative damage from lipids that has been linked to ageing, cancer and many other co-morbidities[70]. The ability of the carnitine shuttle to generate acetyl-CoA is vital for the successful generation of $FADH_2$ and the regeneration of ATP at the end of the electron transport chain. Breakdown of this mechanism is terminal to the cell. We found that a decrease in the levels of several carnitines at higher levels of frailty could be potential indications that general cell-based lipid metabolism is deteriorating, but it is essential that further experimentation needs to be performed to confirm and validate this hypothesis. The importance of the kynurenine pathway provides a conduit for the consumption of over 99% of ingested tryptophan that is unused in protein synthesis[71,72]. With an upregulation of tryptophan noted within the frail metabotype, and with age-related sarcopenia known to be an underlying phenotype within frailty, this observation suggests that muscle protein breakdown is a potential contributor to frailty metabolic output. Further along the kynurenine pathway, a bottleneck in the biogenesis of the vital energy co-factor NAD and its associated dysregulation has also been associated to mitochondrial disturbances[73], activation in times of stress and immune activation[74] alongside links to neurodegenerative diseases[75]. Conversely, the tryptophan kynurenine pathway is also the starting point for the biosynthesis of two related neurotransmitters; serotonin and melatonin. Previous work has indicated that an over activation of this pathway can lead to activation of the immune system and downstream accumulation of potentially neurotoxic intermediates such as quinolinic acid[72] and kynurenic acid[76]. These metabolites are currently considered to be involved in some way in Alzheimer's disease, Parkinson's disease, Huntington's disease and amyotrophic lateral sclerosis and future works in frailty metabolism should consider them as interesting mechanistic targeted[76].

This work exemplifies the high suitability for combined metabolic and pathway analysis to explore and uncover significantly modulated biological pathways within biogerontology. The longitudinal nature of the study, alongside the unselected aspect of the sample cohort are strengths that increases the external validity of the findings. However, several limitations also exist, and these should be considered. While providing results that are consistent with data from previous experimental literature, our findings should be considered hypothesis generating in nature.

This fact, tied to the restricted geography of the cohort (all subjects residing in England), requires that further validation from a range of independent cohorts is essential to test the conserved nature of the results.

Also, to understand fully the complex biological processes that are dysregulated as a component of frailty, a comprehensive systems-based approach is needed to model all dimensions of the process. Further work is needed link metabolic profiles to genotypic expression. Several genetic mutations and markers have already been identified in model organisms[77,78] and humans living extremely long lives[79,80] and these observations need to be related to RNA, protein and metabolite expression. Candidate gene-association studies on data from the same wave of ELSA have indicated genetic changes effecting lipoprotein receptor-related protein 1 (*LRP1*) gene on chromosome 12[81]. This multi ligand receptor has previously been reported to be involved in lipid homeostasis including cholesterol transport; thus, supporting our theory of global lipid imbalance in frailty.

There are also several limitations that need to be considered within the Mendelian Randomization work. Firstly, as frailty is a complex condition and as such is likely to involve multiple genetic variants. The genetic variants typically explain only a small fraction of the total variance in traits; therefore, MR studies

require very large sample sizes for sufficient statistical power. Although we chose the two-sample approach to achieve greater power, our sample size with 1500 cases and 3500 controls may not be powerful enough.

Secondly, the instruments we employed may be considered weak, as indicated by the large confidence intervals of the causal estimates. Finally, for individual polymorphisms the variance explained is usually <1%; therefore, it is advisable to combine multiple polymorphisms into a single allele score to maximize the explanatory power of the instrument. However, due to lack of reported significant SNP-metabolite associations, the available number of genetic instruments was a single or maximum two SNPs for the exposures. The lack of multiple genetic instruments also prevented us from carrying out a pleiotropy test. One of the assumptions of MR analysis is that there is no horizontal pleiotropy, i.e., when a genetic variant affecting multiple traits via separate pathways[82]. The MR-Egger regression method provides valid causal estimates in the presence of some violations of the MR assumptions. However, as this method requires more than two genetic variants assigned to the same exposure, we could not test for this and assumed no pleiotropic effects. Also, in order to obtain more conclusive evidence on the effect of carnitine levels on frailty, studies with sufficiently large sample size are required. While our results should be interpreted with caution, this is an important exercise towards identifying causal relationships.

In summary, our work reveals that the presence of frailty, with an associated increased risk of negative health outcomes in later life, is not only just identifiable through symptomatic presentation but, as predicted by Fried and colleagues[21], is multifactorial and subsequently recognisable by a distinct biochemical phenotype. Our results, primarily imply that a deterioration in lipid metabolism is present within those who clinically present as frail: The downstream set of metabolic observations detected within this study (primarily linked to energy dysregulation) are directly linked with the primary clinical description for frailty: a reduction in physiological reserve. Metabolic frailty measurement has the potential to contribute greatly to the standardisation of frailty assessment. In addition, the application of metabolomics in combination with other -omics based technologies (such as we have with Mendelian Randomization) offers the potential for a greater understanding of the biologic basis and complexity of frailty. Knowledge of frailty risk factors and biomarkers offers the scope to yield effective early stage interventions that can be incorporated into standard of care practices and ultimately contribute to healthy ageing.

## Methods

**Sample collection procedures**. The English Longitudinal Study of Ageing (ELSA) is a continuing cohort study that contains a nationally representative sample of men and women born on or before February 1952 living within England. Data collected at Wave 4 (2008–09) were used as the data source and serum sample source for this study. Data collected at Wave 6 (2012–13) were used for the longitudinal validation of the model (786 samples). This study was performed in compliance with all relevant ethical regulations and guidelines for work with human participants. Participants gave informed written consent to participate in the study and ethical approval was obtained from the London Multi-Centre Research Ethics Committee. Clinical Measurements: Nurses collected anthropometric data (weight, height, waist circumference), blood pressure (BP), and non-fasting blood samples using standard protocols developed within the Health Survey for England[83]. Body weight was measured using Tanita electronic scales without shoes and in light clothing, and height was measured using a Stadiometer with the Frankfort plane in the horizontal position. Body mass index (BMI) was calculated by the standard equation—weight (kilograms)/height (metres) squared. Detailed information on biochemical blood analysis, the internal quality control, and the external quality assessment for the laboratory are summarised in the Supplementary Information.

**Metabolite profiling**. Untargeted metabolite profiling was performed on serum samples that were collected from participants using standard serum collection

techniques[83] and stored at −80 °C prior to analysis. Ultra-High Performance Liquid Chromatography Mass Spectrometry (UHPLC-MS) and gas chromatography mass spectrometry (GC-MS) were performed in tandem on each sample using the Dunn[51] and Begley[84] protocols with some minor alterations and is briefly summarised as follows:

Metabolites were extracted from the 1191 serum samples from Waves 4 and 100 samples from Wave 6 by individually adding 900 μL of an organic solvent mixture of 80% methanol/15 water/5% acetonitrile to 330 μL of serum. Subsequent vortexing and centrifugation (17,500 × g) yielded a metabolite rich supernatant that was split in to two aliquots and lyophilised for 12 h to yield a metabolite pellet that was stored at −80 °C prior to analysis. A pooled QC standard was also generated by combining 30 μL aliquots of each sample in to a pooled vial with subsequent 330 μL portions being extracted identical to each sample.

Processed metabolite pellets were defrosted at 4 °C and subsequently reconstituted in 100 μL of mobile phase A. UHPLC-MS analysis was performed using an Accela UHPLC auto sampler system coupled to an electrospray LTQ-Orbitrap XL hybrid mass spectrometer (ThermoFisher, Bremen, Germany). Analysis was carried out in positive ESI mode while samples in each run were completely randomised to negate for any bias. A gradient type UHPLC method was used during each run with 95% water/5% methanol/0.1% formic acid as mobile phase A and 95% water/5% methanol/0.1% formic acid as mobile phase B. 5 μL of the extract was injected onto a Hypersil GOLD UHPLC $C_{18}$ column (length 100 mm, diameter 2.1 mm, particle size 1.9 μm, Thermo-Fisher Ltd. Hemel Hempsted, UK) held at a constant temperature of 50 °C while a solvent flow rate of 400 μL min$^{-1}$ was used to drive the chromatographic separation.

Mass calibration was carried out in accordance with the manufacturer's guidelines using caffeine (20 μg mL$^{-1}$), the tetrapeptide MRFA (1 μg ml$^{-1}$) and Ultramark 1621 (0.001%) in an aqueous solution of acetonitrile (50%), methanol (25%) and acetic acid (1%). Acquisition settings for initial profiling were carried out at 30,000 resolution in centroid and ran at 1 μ-scan per 400 ms in the 100–1000 $m/z$ range with source gases set at sheath gas = 40 arbitrary units, aux gas = 0 arbitrary units, sweep gas = 5 arbitrary units. The ESI source voltage was set to 3.5 V, and capillary ion transfer tube temperature set at 275 °C.

Xcaliber software Version 3.0 (Thermo-Fisher Ltd. Hemel Hempsted, U.K.) was used as the operating system for the Thermo LTQ-Orbitrap XL MS system. Data processing was initiated by the conversion of the standard UHPLC raw files in to the NetCDF format via the software conversion tool within Xcaliber.

Peak picking was carried out in R-Studio (www.rstudio.com) using the XCMS algorithm (http://masspec.scripps.edu/xcms/xcms.php). The output yielded a data matrix of mass spectral features with related accurate $m/z$ and retention time pairs. Data from the internally pooled QC samples were then used to align for instrument drift and quality control (via application of an in-house robust spline alignment Matlab script). The data matrix was also signal corrected to remove peaks that crossed the 20% RSD threshold within QC samples across the analytical run.

GCMS Analysis was carried out on a Leco Pegasus 3 Time-of-Flight mass spectrometer coupled to an Agilent 6890 GC oven and Gerstel-MPS autosampler. Derivatization and instrument conditions were identical to those used by the Begley protocol[84] to yield raw data files. These were subsequently converted in to NetCDF files within chromaTOF acquisition software. Peak picking was carried out in R-Studio using the XCMS algorithm (http://masspec.scripps.edu/xcms/xcms.php). The output yielded a data matrix similar to the retention time and quant mass values contained within an internal GC standard library containing over 1600 pure reference compounds run under identical conditions.

All metabolites identified as significant within the analysis were assessed and scored according to rules set out by the Chemical Analysis Working Group of the Metabolite Standards Initiative[41]. Where available, pure reference standards were purchased (Sigma-Aldrich, St Louis, USA) and used to confirm the highest level of metabolite identification—Level 1. Where no standard was available, matching of measured MS/MS spectra against those from within the METLIN metabolite database (http://metlin.scripps.edu/) was performed to give a Level 2 annotation confirmed by appropriate secondary ion $m/z$ values. All scoring is available in Supplementary Table 1.

UHPLCMS data dependent MS$^n$ analysis was performed on chemical standards using a LTQ-Orbitrap XL hybrid mass spectrometer (ThermoFisher, Bremen, Germany). Precursor ion full scan was performed followed by an additional scan where the ion of interest for trapped within the linear ion trap for 1000 ms and subsequently subjected to CID of 50 au, following which the fragment ions were detected. A minimum of three scans were recorded for both precursor and product ions. A combined spectrum of both FTMS and ion-trap data was used to generate product ion lists and intensity. The same tuning method, injection volume, CID and activation energy were applied to QC and standard sample to standardise the comparison. For all metabolites analysed, retention time was matched within 20 s or below (extracted date is highlighted in Supplementary Figs. 16–30.

All metabolites identified via GCMS were retention time and fragmentation matched to an internal standard library that was analysed under identical conditions as to the main analysis.

**Chemometrics**. PC-DFA, RUSBoost-CART and robust spline alignment analysis were carried out using MATLAB 2012a (MathWorks, Natick, MA, USA). Prior to chemometric analysis data matrices were log$_2$-transformed to account for skewed

distribution. All tests were supervised and bootstrapped (×10,000) using groups determined by FI value. In this process, each data set was split in to training/tests sets and resampled[85]. Sex stratified PC-DFA plots are also documented in Supplementary Fig. 31 (Wave 4) and Supplementary Fig. 32 (Wave 6). Linear regressions models and prediction plots were performed using the lm-function in R-Studio (Version 1.0.44). Univariate $t$-tests, cross validation, heat-map correlation curves and ROC curves were performed using MetaboAnalyst 3.0[86] http://www.metaboanalyst.ca/. Due to the unbalanced nature of the sample classes (non-frail vs. frail) a by Monte-Carlo cross validation (MCCV) was use to balance the groups. In each MCCV, two thirds (2/3) of the samples were used to evaluate the feature importance. The top 12 (Wave 4) and top 11 (Wave 6) important features were then used to build classification models which were validated on the 1/3 samples that were left out. The procedure was repeated 500 times to calculate the performance and confidence interval of each model. Classification and feature ranking were performed using aPLS-DA algorithms using seven latent variables as input to determine the final ROC curves.

**Network analysis**. Mummichog (Version 1.0.5) pathway analysis[34] was used offline in Python (Version 3.5.2) to predict network activity from pre-processed UHPLC-MS metabolomics data. $[M + H]^+$ was selected as the force primary ion ($z$) alongside an evidence cut-off score of 3 to include a metabolite within an activity network ($e$). The full metabolite data set was used as an input and 554 extracted features were determined as significant ($p < 0.0001$) from the associated t-test yielding 263 potential metabolites. From this, 22 network modules were generated using an activity network of 23 annotated and statistically significant metabolites. Output files were visualised in Cytoscape (Version 3.4.0) (Supplementary Fig. 13) where manual addition of GCMS data was performed to generate the enriched hybrid frailty model (Fig. 4). Metabolite non-frail to frail distribution are available in Supplementary Figs. 33 and 34.

**Mendelian randomization**. A text-mining approach using the keyword 'carnitine' yielded four possible exposures and five genetic instruments (single nucleotide polymorphisms—SNPs). Instruments were assigned into the same exposure if the reported direction of effect and the study[43,44,87] were the same.

Exposure 1: Blood metabolite levels (unit increase) (carnitine), SNPs: rs12356193, rs419291[43].

Exposure 2: Blood metabolite levels (unit decrease) (carnitine), SNP: rs1466788[43].

Exposure 3: Acylcarnitine levels (unit increase) (Carnitine), SNP: rs1171606[44].

Exposure 4: Metabolic traits (unit decrease) (carnitine), SNP: rs7094971[87].

Rs7094971 (Exposure 4) was excluded from further analyses, as it was in high linkage disequilibrium with rs12356193 ($r2 = 0.87$). For Exposure 2 (rs1466788) and Exposure 3 (rs1171606) the direction of association for the outcome and the exposure was the same, against the expectations. MR analysis results of causal estimates are summarised in Supplementary Table 2.

**Reporting summary**. Further information on research design is available in the Nature Research Reporting Summary linked to this article.

## Data availability

All metadata, mass spectrum files and statistical packages used in this paper are freely available and deposited in accessible public repositories. All English Longitudinal Study of Ageing (ELSA) data files are available from the United Kingdom Data Service repository—Study Number 5050 (http://discover.ukdataservice.ac.uk/catalogue?sn=5050). Mass Spectrum and metabolomics data are accessible through the EMBL-EBI MetaboLights repository—Study Identifier MTBLS598 (www.ebi.ac.uk/metabolights/). Statistical scripts used to perform PC-DFA, PLS-R and PLS-DA were developed within the www.biospec.net cluster-toolbox and are freely available on the open source GitHub repository hosted at github.com/Biospec/cluster-toolbox-v2.0. The source data underlying Figs. 1–5 and Supplementary Figs. 1–15 and 31–34 are provided within the supplied Source Data file alongside data used to generate the Z-scores. Supplementary Figs. 16–30 were generated in the Thermo Fisher Xcaliber Software using the raw LCMS data available within the upload supplied to the MetaboLights repository. All data are available from the corresponding author upon reasonable request.

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

## Acknowledgements

The fRaill Project is funded by the UK Medical Research Council (190 MRC G1001375/1) and the authors would like to thank Linda Wilson and Margaret Blake at NatCen (www.natcen.ac.uk/) for help with sample logistics. N.J.W.R. would also like to express thanks to Dr Ying Chen from Yale School of Public Health alongside the Strathclyde Ageing Network for constructive discussions.

## Author contributions

N.J.W.R.: Conception and design of work, data collection, data analysis and interpretation, drafting and editing of article. D.K.T.: Conception and design of work, data collection, data analysis. Y.X.: Conception and design of work, data analysis. I.R.W. and Z.R.: Data collection and drafting/editing of article. C.H.J.: Data analysis, drafting, and editing of article. K.M.: Conception and design of work, data collection, data analysis, editing of article. B.V. and A.D.M.: Conception and design of work, data collection, data analysis. G.T.: Conception and design of work, data collection and data analysis. F.C.W.W. and T.C.: Conception and design of work, interpretation of results and editing of article. N.P., J.N. and R.G.: Conception and design of work, supervision of data analysis, interpretation of results and editing of article.

## Competing interests

The authors declare no competing interests.
