## [Peer Review File · Nature Communications]

Reviewers' comments:

Reviewer #1 (Remarks to the Author):

I would like to acknowledge the authors for their further explanations and comments.

However, basically nothing has changed and, in my opinion, the key issue (as agreed by the authors), particularly for a top journal, would be independent replication. That is not available and there are quite many bothering issues related to the internal replication attempt on Wave 6 data in the supplement (eg the statistical assessments of these comparisons are totally missing; however the figures per se do not give a strong argument on the strength of the results). I am not convinced based on these data.

In addition, I have been commenting all the time that the biological elaboration of the (non-replicated) results goes a lot too far being in many senses more speculation than based on solid biological results. The response to this from the authors is "we recognise the need for replication in an independent cohort" – this is not the solution for the current extensive speculations.

Thereby, I am sorry to say, I keep remaining very critical for the robustness (and replicability) of these results and the extensive biological interpretations given.

Reviewer #3 (Remarks to the Author):

No additional points.

Reviewer #1:

Reviewers' comments:

I would like to acknowledge the authors for their further explanations and comments.

However, basically nothing has changed and, in my opinion, the key issue (as agreed by the authors), particularly for a top journal, would be independent replication. That is not available and there are quite many bothering issues related to the internal replication attempt on Wave 6 data in the supplement (eg the statistical assessments of these comparisons are totally missing; however the figures per se do not give a strong argument on the strength of the results). I am not convinced based on these data.

In addition, I have been commenting all the time that the biological elaboration of the (non-replicated) results goes a lot too far being in many senses more speculation than based on solid biological results. The response to this from the authors is “we recognise the need for replication in an independent cohort” – this is not the solution for the current extensive speculations.

Thereby, I am sorry to say, I keep remaining very critical for the robustness (and replicability) of these results and the extensive biological interpretations given.

Reviewer #1 Response

We appreciate the continued input from Reviewer #1 and to address the highlighted concerns we have substantially changed two main aspects of the paper that hopefully work to this end and clarify the robustness of our observations. As we understand it these are:

- *The lack of statistics behind the metabolic model - significance and how it can predict frailty alongside similar signal within the internal replication sample set.*

To this end we have now performed extensive descriptive statistical analysis and identified a panel of 12 metabolites that are significant ($P < 0.05$, FDR corrected) in differentiating non-frail and frail phenotypes. These metabolites have subsequently been used to generate a multivariate-ROC model (with associated confidence intervals) that acts as a tool to describe the diagnostic ability of frailty prediction. The results (**fig 5** and **supplementary fig 6**) from this indicate a highly significant model, especially considering the sample size and number of variables used within the model. ***From a serum sample, the selected metabolite panel can predict if a person clinically presents as frail.*** This has never been achieved before and is a result of the highest significance. We have subsequently replicated this same multivariate-ROC model within the Wave 6 replication set. We get the same result, even though three of the metabolites are not present in the Wave 6 data (due to detection limits and samples size – we fully know we need to independently replicate but this also shows the strength of the combined signals that are in the model).

- *Over interpretation of extensive biological results*

We have accepted that within the conclusion we associate the carnitine metabolites and vitamin E to act in a synergistic manner, with no real empirical new data for this – only evidence from literature alongside the identification of the metabolite panel, that to be fair, are linked within specific biochemical pathways. This is a fair comment from the reviewer and we have subsequently gone through our whole article and toned down this speculation and other similar assumptions.

Reviewer #3 (Remarks to the Author):

No additional points.

Reviewer #3 Response

N/A

REVIEWERS' COMMENTS:

Reviewer #1 (Remarks to the Author):

I would like to acknowledge the authors again for their further explanations, comments and changes in the manuscript. The work has again improved. The presented new and partly replicated ROC analyses are interesting. However, they do not provide diagnostic performance (AUC of 0.755) as indicated by the authors in the abstract.

NCOMMS-19-00967-T Final Response to Referee's Questions

"Metabolic dysregulation in vitamin E and carnitine shuttle energy mechanisms identified as drivers behind human frailty"

Nicholas J W Rattray, Drupad K Trivedi, Yun Xu, Tarani Chandola, Caroline H Johnson, Alan D Marshall, Kris Mekli, Zahra Rattray, Gindo Tampubolon, Bram Vanhoutte, Iain R White, Frederick C W Wu, Neil Pendleton, James Nazroo & Royston Goodacre

Reviewer #1:

I would like to acknowledge the authors again for their further explanations, comments and changes in the manuscript. The work has again improved. The presented new and partly replicated ROC analyses are interesting. However, they do not provide diagnostic performance (AUC of 0.755) as indicated by the authors in the abstract.

The authors appreciate the further input from Referee #1 – your input has helped shape the article and added to the depth of research and understanding within. With regards the mROC result of 0.755 we refer to paper by Swets¹. Within it the authors offer benchmarks for gauging AUCs, suggesting that values ≥ 0.9 are “excellent,” ≥ 0.80 “good,” ≥ 0.70 “fair,” and < 0.70 “poor.” Within the text we refer to the mROC diagnostic performance as “a moderately strong level of performance”. We are not claiming the prediction is excellent but a level of diagnostic ability is present within the system and we have documented it as such. But to placate this comment we have removed the statement that refers to the biomarker panel explicitly acting as a predictor of frailty’

References

1. Swets J A, Dawes R M and Monahan J. “Psychological science can improve diagnostic decisions.” *Psychological Science in the Public Interest*. 1, 1–26 (2000).